# Developmental Strategy and Validation of the Midwifery Interventions Classification (MIC): A Delphi Study Protocol and Results from the Developmental Phase

**DOI:** 10.3390/healthcare11060919

**Published:** 2023-03-22

**Authors:** Giulia Maga, Cristina Arrigoni, Lia Brigante, Rosaria Cappadona, Rosario Caruso, Marina Alice Sylvia Daniele, Elsa Del Bo, Chiara Ogliari, Arianna Magon

**Affiliations:** 1Department of Biomedicine and Prevention, University of Rome Tor Vergata, 00133 Rome, Italy; 2Department of Public Health, Experimental and Forensic Medicine, Section of Hygiene, University of Pavia, 27100 Pavia, Italy; 3Department of Women’s and Children’s Health, Faculty of Life Sciences and Medicine, King’s College London, London WC2R 2LS, UK; 4Department of Medical Sciences, University of Ferrara, 44121 Ferrara, Italy; 5Health Professions Research and Development Unit, IRCCS Policlinico San Donato, San Donato Milanese, 20097 Milano, Italy; 6Department of Biomedical Sciences for Health, University of Milan, 20133 Milan, Italy; 7Department of Midwifery and Radiography, School of Health and Psychological Sciences, University of London, London EC1V 0HB, UK; 8Department of Clinical-Surgical, Diagnostic and Pediatric Sciences, University of Pavia, 27100 Pavia, Italy

**Keywords:** delphi, protocol, maternal and newborn health, midwifery, quality indicators

## Abstract

This study protocol aims to describe the rationale and developmental strategy of the first study in the Italian context which aimed to define a Midwifery Interventions Classification, an evidence-based, standardized taxonomy and classification of midwifery interventions. Midwifery interventions require a specific definition, developed through a consensus-building process by stakeholders to develop the Italian taxonomy of the Midwifery Interventions Classification with the potential for international transferability, implementation, and scaling up. A multi-round Delphi study was designed between June and September 2022, and data collection is planned between February 2023 and February 2024. The developmental phase of the study is based on a literature review to select meaningful midwifery interventions from the international literature, aiming to identify an evidence-based list of midwifery interventions. This phase led to including 16 articles derived from a systematic search performed on PubMed, CINAHL, and Scopus; 164 midwifery interventions were selected from the data extraction performed on the 16 included articles. Healthcare professionals, researchers, and service users will be eligible panelists for the Delphi surveys. The protocol designed a dynamic number of consultation rounds based on the ratings and interim analysis. A nine-point Likert scoring system is designed to evaluate midwifery interventions. Attrition and attrition bias will be evaluated. The results from the study designed in this protocol will inform the development of the Italian taxonomy of the Midwifery Interventions Classification. A shared classification of midwifery interventions will support audit and quality improvement, education, and comparable data collections for research, sustaining public recognition of midwifery interventions to promote optimal maternal and newborn health.

## 1. Introduction

Maternal and neonatal care quality indicators usually relate to disease or adverse event prevention and seldom to health promotion [1,2]. Hence, there is the potential to define healthcare indicators in line with current midwifery standard care practice based on a salutogenic approach to maternity care, which is mainly unmet for specific practice areas [1,2]. Some projects led to the definition of specific midwifery core outcome sets, such as the Italian Midwifery Core Outcomes Set (M-COS), which was recently developed to evaluate midwifery core outcomes underpinned by a salutogenic framework [3]. In the Italian context, although the M-COS was developed, midwifery interventions still remain without specific guidance based on a methodological development study to align the views of field experts in defining an Italian taxonomy of the Midwifery Interventions Classification (MIC). Even if classifications, such as M-COS or MIC, have to be context-specific, they are relevant to the international scientific arena, as classifications might stimulate international debate and the need to further align context-specific classifications internationally.

### Background

Quality of care is a core issue of maternal and newborn health [2,4,5,6]. The evidence shows that care provided by midwives is associated with improved health outcomes for mothers and newborns [7]. Donabedian A. et al. (1966) defined the quality of care as the extent to which actual care conforms with the present criteria for good care [8]. Thus, to define the criteria for good care, it is necessary to consider three indicators described by the Donabedian framework: structure, process, and outcomes [9]. The structure includes variables related to the healthcare professionals, such as experience and skills; patients, how old they are and their state of health; and organization, as an organic endowment and care model. The process refers to care interventions; these can be planned and carried out independently, following a medical prescription, or by collaborating with a multi-disciplinary team. Finally, outcomes correspond to the health outcomes of patients related to the care provided.

Most of the available quality indicators in maternal and neonatal care typically refer to illness or adverse event prevention, with less attention to health promotion [10,11]. Therefore, there is an increased awareness of the need for quality indicators following the current midwifery standard care practices, focused on promoting a salutogenic approach to maternity care [11,12]. In this perspective, the Italian Midwifery Core Outcomes Set (M-COS) was recently developed to evaluate midwifery core outcomes sensitive to a salutogenic framework [3]. Even though the core outcomes have been identified, it is still unknown which midwifery interventions are likely to benefit each core outcome. Standardizing and classifying healthcare into datasets allows documentation, communication, sharing of data across systems, evaluating outcomes, conducting effectiveness-related research, measuring productivity, assessing competencies, facilitating reimbursement, and determining staffing needs [13].

Midwifery interventions are defined as elements of maternity care provided by midwives to improve and optimize the health outcomes of women, newborns, and the public health of society at large. The Lancet’s Series on Midwifery developed the Quality Maternal and Newborn Care framework [14] that describes the full scope of midwifery care that should be accessible to all women and newborns. The framework identifies some essential components of midwifery care that are interlinked, such as effective practices, the organization of care, the philosophy and values of the care providers working in the health system, and the characteristics of care providers. However, there is a lack of consensus on the core midwifery interventions [14]. The literature highlights a wide heterogeneity of maternity care interventions provided by different health professionals, such as midwives, nurse-midwives, obstetrics, and physicians [11,15,16].

In light of the above consideration, identification and consensus from key stakeholders on core interventions are pivotal to ensuring quality midwifery care. Each midwifery intervention will be listed with a label name and a definition and classified into a domain and a class. The scope of the MIC will cover healthy women with low-risk pregnancies and their newborns, from antenatal to birth and postnatal care. The reference frameworks will be the Donabedian framework [9], the salutogenic framework [17], and the Quality Maternal and Newborn Care framework [14]. This protocol aims to describe the developmental strategy to define a Midwifery Interventions Classification (MIC), an evidence-based, standardized taxonomy and classification of midwifery interventions based on the Italian maternity care context.

## 2. Materials and Methods

### 2.1. Design

This protocol was designed including a multi-method and multi-phase approach between June and September 2022, consistent with the recommendations of The COMET Handbook: version 1.0 [18] and the Guidance on Conducting and REporting DElphi Studies (CREDES) [19]. The study will be conducted between February 2023 and February 2024. This protocol is part of a more comprehensive line of research registered with the Core Outcome Measures in Effectiveness Trials (COMET) Initiative (registration number 1723; available online at https://comet-initiative.org/Studies/Details/1723, accessed on 1 March 2023).

The study will consist of two stages. The first stage (Developmental phase) was recently completed and included conducting a literature review to develop a preliminary and evidence-based version of the MIC; the second stage (Consensus process) will validate the classification through a Delphi survey involving different stakeholders: healthcare professionals, healthcare researchers, and service users. Midwifery interventions resulting from Stage 2 will be taxonomized into domains and classes constituting the final version of the MIC. Figure 1 summarizes the study design.

#### 2.1.1. Stage 1: Developmental Phase

Stage 1 aimed to identify an evidence-based list of midwifery interventions and develop a preliminary version of the Midwifery Interventions Classification. A literature review was performed following the Preferred Reporting Items for Systematic Reviews and Meta-Analysis (PRISMA) statement and flowchart [20]. We updated the literature review conducted previously for the M-COS development by searching three primary databases: PubMed, CINAHL, and Scopus, using the exact keywords used once, such as maternal and newborn health, midwifery, and quality indicators. Moreover, we hand-searched the bibliography of relevant papers to identify any additional references. Studies selection was based on the following inclusion criteria: (a) papers in the English or Italian language with full-text availability without time limits; (b) all types of study design; (c) studies describing midwifery interventions during pregnancy, childbirth, and the postnatal period in any care setting; and (d) studies conceived within a physiological framework of maternity care. Then, midwifery interventions identified through the literature review were listed, constituting the preliminary version of the Midwifery Interventions Classification (MIC).

#### 2.1.2. Stage 2: Consensus Process

Stage 2 will aim to achieve consensus on MIC through a Delphi technique, an established method for reaching consensus among field experts and stakeholders [21]. Expressly, we will use the Delphi technique to converge opinions from stakeholders regarding the importance of different midwifery interventions in sequential questionnaires sent electronically. To better manage this process, we will use a web-based system [18]. Midwifery interventions resulting from Stage 2 will be taxonomized into domains and classes constituting the final version of the MIC.

### 2.2. Participants: Panel Composition

In the midwifery and maternity context, different stakeholders’ opinions are expected. We will involve three diverse stakeholder groups, each forming a balanced skill-mixed panel: healthcare professionals, healthcare researchers, and service users. The healthcare professionals panel will comprise midwives with at least one year of experience in midwifery care. The panel of healthcare researchers will involve experts on the Delphi technique and its methodology. The third panel will include healthy women who experienced a physiological pregnancy and birth within the last five years. We will use diverse Participant Information Sheets using different terminology depending on the stakeholder’s group.

#### Group Size

Group size is a pragmatic choice in a Delphi process, not based on statistical power [18]. Otherwise, each panel should have a good participant representation, with qualified experts who deeply understand the issues [21]. Previous obstetric studies using the Delphi technique have considered a group size of around 20–40 participants for patients and 50–100 for healthcare professionals [22].

### 2.3. Data Collection and Analysis

#### 2.3.1. Number of Rounds

The number of rounds in a Delphi can be dynamic, but there will be a point beyond which a greater degree of consensus is unnecessary or unlikely to be achieved [18]. Therefore, we planned two rounds with the option of adding a third if further prioritization is warranted. In Round 1, participants will be asked to assess the importance of each midwifery intervention, with the possibility of adding new ones if deemed appropriate. In Round 2, participants will receive statistical feedback on the previous round’s results; hence, they can reconsider their judgment and assess new interventions added (if any).

#### 2.3.2. Structure of the Questionnaire

Designing the Delphi questionnaire requires careful consideration. Different studies investigated the “consistency effect”, for which participants are influenced to answer items by the previous ones [18]. Thus, we will follow the recommendations that general questions should precede specific ones and be grouped into topics. Moreover, the preliminary version of the MIC may not be entirely exhaustive; then, we will include an open question at the end of the Round 1 questionnaire to identify additional midwifery interventions.

#### 2.3.3. Scoring System

We will use the 9-point Likert scoring system to evaluate midwifery interventions, as recommended by the Grading of Recommendations Assessment, Development and Evaluation Working Group [23]. Expressly, 1 to 3 signifies a midwifery intervention is of little significance, 4 to 6 essential but not critical, and 7 to 9 crucial. Furthermore, we will include an “unable to score” category to allow for the fact that some participants may not have the level of expertise to rate specific midwifery interventions.

#### 2.3.4. Feedback between Rounds

There is no evidence of the impact on the final list of items of retaining or dropping items between rounds [18]. On one hand, there is a more holistic approach to maintaining all items between rounds, enabling participants to score and prioritize the list of items as a whole; on the other, the option to reduce participants’ burden of having an extensive initial list. Therefore, we will follow an intermediate approach: to retain all midwifery interventions between rounds 1 and 2, enabling participants to re-score in light of feedback for every item and then drop items in subsequent rounds. We will present results for each midwifery intervention aggregated across stakeholder groups through descriptive statistics at the end of each round.

#### 2.3.5. Consensus Definition

To define consensus, we will follow the “70/15%” method [18] implemented by Wylde et al. [24]. This approach requires evaluating more than 70% of participants scoring 7–9 and less than 15%, scoring 1–3 for maintaining midwifery interventions between rounds [18]. Moreover, we will add the additional criterion introduced by Wylde et al. (2015): midwifery interventions scored as 7–9 by more than 90% of members of one panel will also be carried forward to the subsequent round, regardless of the ratings of the other panel [24]. The rationale is to include midwifery interventions deemed crucial by most participants and ensure that midwifery interventions considered exceptionally important by only one panel will not be omitted.

### 2.4. Ethical Considerations

All input and involvement during the process will remain highly confidential. Based on the information given, each invited participant will either agree to participate in the study or exercise their right to withdraw at any time. Data will be managed following the General Data Protection Regulation (EU GDPR).

### 2.5. Validity and Reliability

The assessment of the validity and reliability of the results derived from this study will reflect the strategy to manage the attrition and attrition bias between rounds. Attrition is the degree of non-response after the first round of the Delphi that can vary widely between studies. It may depend on many factors, e.g., the timing of Delphi rounds, the length of the Delphi, and the method of recruitment of participants. Although there is no guidance on what constitutes an acceptable response rate, The COMET Handbook suggests that around 80% for each stakeholder group is deemed acceptable [18]. Attention must be paid to the attrition bias occurring when the participants that do not respond in subsequent rounds have different views from their stakeholder group peers who continue to participate [18]. We will evaluate whether attrition will introduce bias between rounds by comparing average scores for those completing round 2 and those dropping out after round 1, and do the same for subsequent rounds [25].

## 3. Preliminary Results: Stage 1—Developmental Phase

From a previously published literature review, we identified two included studies out of eleven that were consistent with the need to identify midwifery interventions. We updated the literature review to develop the preliminary list of MIC for the future phases of this study. The search strategy is described in Table 1, and the searches were updated until 15 February 2023. The search identified 338 records, from which we removed 35 duplicates. In addition, two other duplicates were removed from the current selection flow because they were included directly as pertinent from the previously published literature review. After screening the remaining 301 records, we found that 211 were not focused on midwifery interventions, resulting in 90 articles being retrieved. Of these 90 articles, 64 were not relevant to our study, while 4 studies focused solely on maternal or neonatal diseases. We also found 10 studies that were part of the same project and had identical midwifery interventions, leading us to include only the project’s first study. This decision was made because the content of the other 10 studies did not differ regarding midwifery interventions. Therefore, we included the first study only once to extract data relevant to our search. In addition, two articles were excluded because they were not published in a language accessible to the authors (i.e., English or Italian). Therefore, the eligible articles were ten identified from the queries and four records from the manual search of the references available in the eligible records. The 14 eligible articles were included after the full-text evaluation and were added to the two articles included in the flow from the previous literature review. Figure 2 depicts the selection flow.

Appendix A provides the overview of the data extraction for generating a preliminary MIC from the 16 included studies [26,27,28,29,30,31,32,33,34,35,36,37,38,39,40,41] by extracting the geographic area where the study was performed, the study design and objective, and a taxonomy including the domain (broader topics), classes (intermediate topics), and midwifery interventions (specific topics). The included articles encompassed 11 observational studies [27,28,29,30,32,33,34,35,39,40,41] and 5 literature reviews [26,31,36,37,38]. Figure 3 shows the distribution of the countries where the studies were performed, and Table 2 summarizes the preliminary version of the MIC derived from Appendix A (Overview of the data extraction for generating a preliminary MIC from the literature review) after extracting, translating, merging, and retitling interventions.

## 4. Discussion of the Preliminary Results from the Stage 1—Developmental Phase

Stage 1 of this study outlines a preliminary classification of midwifery interventions across a range of categories and classes that are consistent with previous international classifications [42]. The emerging MIC proposal is important because it helps to organize and standardize the range of interventions that midwives may provide across different contexts and populations, making it easier to track and evaluate the impact of midwifery care on maternal and newborn health outcomes.

By breaking down midwifery interventions into specific categories/domains and classes, this proposed classification also provides a more nuanced understanding of the various factors that contribute to maternal and newborn health. For example, the section on maternal outcomes identifies a range of physiological and behavioral interventions that may impact a woman’s physical and psychological well-being from antenatal to birth and postnatal periods. Similarly, the newborn care section outlines interventions for healthy and sick babies, recognizing that different newborns may require different levels of care and support. The emerging MIC also highlights the importance of interdisciplinary collaboration and coordination across different healthcare settings and providers. For example, the inclusion of birth center/primary care and hospital/physician care categories underscores the need for midwives to work closely with other healthcare providers to ensure seamless continuity of care for women and newborns.

The proposed preliminary version of MIC derived from Stage 1 (Table 2) will serve as a valuable starting point for the development of a clear classification for Italian midwives. This classification can be used as a reference point to identify and prioritize the most relevant interventions for midwifery practice in the Italian context and to guide the development of a consensus-based set of core interventions using a Delphi survey approach, paving the way for international consensus. This can help to ensure that midwives are equipped with the most relevant and evidence-based interventions to provide high-quality care to women and newborns in Italy.

In Italy, midwifery is an established, autonomous profession and self-regulated. A self-regulated profession is responsible for setting and enforcing its own standards of practice, conduct, and ethics [43]. This means that the profession establishes its own rules and regulations for the education, training, and licensure of its members, as well as for the ongoing monitoring and evaluation of their professional performance. Midwives work independently and collaboratively with gynecologists and other healthcare professionals to provide care and support to women from antenatal to birth and postnatal periods. As per the international context, in Italy, midwives promote maternal and infant health, prevent complications, and manage normal labor and birth [44]. In recent years, midwifery has gained increasing recognition and importance in Italy, with a growing emphasis on the role of midwives in providing safe and high-quality care to women and newborns [44]. However, there are still challenges and areas for improvement in the midwifery field, including the need for standardized quality indicators to measure and improve the quality of care provided by midwives. The proposed Delphi process aims to address this need by developing a standardized classification of midwifery interventions that are tailored to the unique needs and contexts of midwifery practice in Italy.

Standardized classifications provide a valuable tool for organizing and standardizing health information and practices, particularly in the context of global health initiatives and efforts to improve healthcare quality and outcomes [45]. However, it is important to recognize that standardized classifications need to be developed and adapted for country-specific use, even if they are based on the international literature and guidelines [46]. While standardized classifications based on the international literature could provide a helpful starting point for developing country-specific classifications, it is important to consider each country’s unique cultural, social, economic, regulative, and political contexts. These factors can influence the relevance and applicability of different interventions and practices and affect the feasibility and sustainability of implementing standardized classifications. In the case of the current study, while the MIC derived from Stage 1 of the research is based on the international literature, it will need to be evaluated and adapted for country-specific use in Italy. This approach will involve engaging with a panel of Italian stakeholders and experts to evaluate, using a Delphi approach, the classification and identification of any interventions or domains that may be more or less relevant to midwifery practice in Italy.

In other words, a country-specific proposal will be developed as the result of the implementation of the next phases of the protocol to ensure that the MIC is relevant and applicable to the unique needs and contexts of midwifery practice in Italy. This tailored approach will enhance the classification’s effectiveness and ensure alignment with the priorities and perspectives of Italian midwives and maternal and newborn health stakeholders.

## 5. Expected Results from the Delphi Study

Having an MIC is highly relevant to sustaining health promotion and salutogenic interventions in maternal and neonatal care in real-world settings [47]. The current study protocol addresses the gaps presented by the current unavailability of an MIC in the Italian context and might also be relevant to boosting an international debate, given the current paucity of international classification in this regard [3]. Even if the MIC derived from this protocol will be Italian context-specific, it has the potential for international scaling up, especially acknowledging that the developmental phase of the study is based on a literature review aimed at retrieving an evidence-based list of midwifery interventions to develop a preliminary version of the MIC. We have planned the work, but the project has yet to start. To date, we have conducted an initial literature review that allowed us to understand the knowledge gap, but that will be updated with Stage 1 of the study.

Taxonomies are also relevant for facilitating training, education programs, and assessments, because they provide a defined scheme of classification and precisely hierarchical classification of interventions within given domains for educators, students, and healthcare workers [48]. Other advantages of developing an MIC also rely on its possibility to be applied in clinical documentation and boost subsequent possible electronically generated data [49,50]. Furthermore, having a clear MIC of reference in a specific context or even internationally might boost the social recognition of midwives because the worth of the delivered maternal and neonatal care might become tangible in the developed classification and more readily understandable from the general public perspective [10,11].

The proposed Delphi approach enables researchers to gather expert opinions through a series of iterative questionnaires to reach a group consensus on defining the MIC. This approach mitigates issues often associated with traditional consensus meetings by anonymity. For instance, anonymity helps reduce the impact of dominant personalities in the debate and decreases peer pressure [51]. In this sense, responses from the experts are weighted equally, and the dynamicity of the feedback process allows experts to mature a precise idea across the required several rounds, encouraging them to reassess their ratings of the previous iterations [52].

### Limitations

The main limitations of the protocol are consistent with the employed methodology. In fact, the process might be perceived by participants to be time-consuming. In this regard, adopting a web-based system should help researchers in optimizing the response rate (Williamson et al., 2017). Other implicit limitations of this approach stem from a restriction of the possibility of elaborating in-depth on the views of the experts as the method does not encompass open discussions. Finally, reaching a consensus does not necessarily imply achieving the best possible MIC. In this sense, future further validation studies (e.g., criterion-related validity studies) are needed to corroborate the MIC in relation to measurable patient-level outcomes and to the previously developed M-COS.

## 6. Conclusions

The MIC is pivotal for highlighting salutogenic midwifery interventions in promoting and maintaining maternal and neonatal health. This protocol encompasses the methodological guidance for defining an evidence-based, standardized taxonomy and classification of midwifery interventions based on the Italian context and with possible international scaling up. The MIC has the potential to facilitate education, be used in clinical documentation, boost comparable gathering of data for research purposes, audit and quality improvement, and promote the understanding of the maternal and neonatal in the general public. Future research will be necessary to corroborate the validation of the results of the study designed in this protocol.

## Figures and Tables

**Figure 1 healthcare-11-00919-f001:**
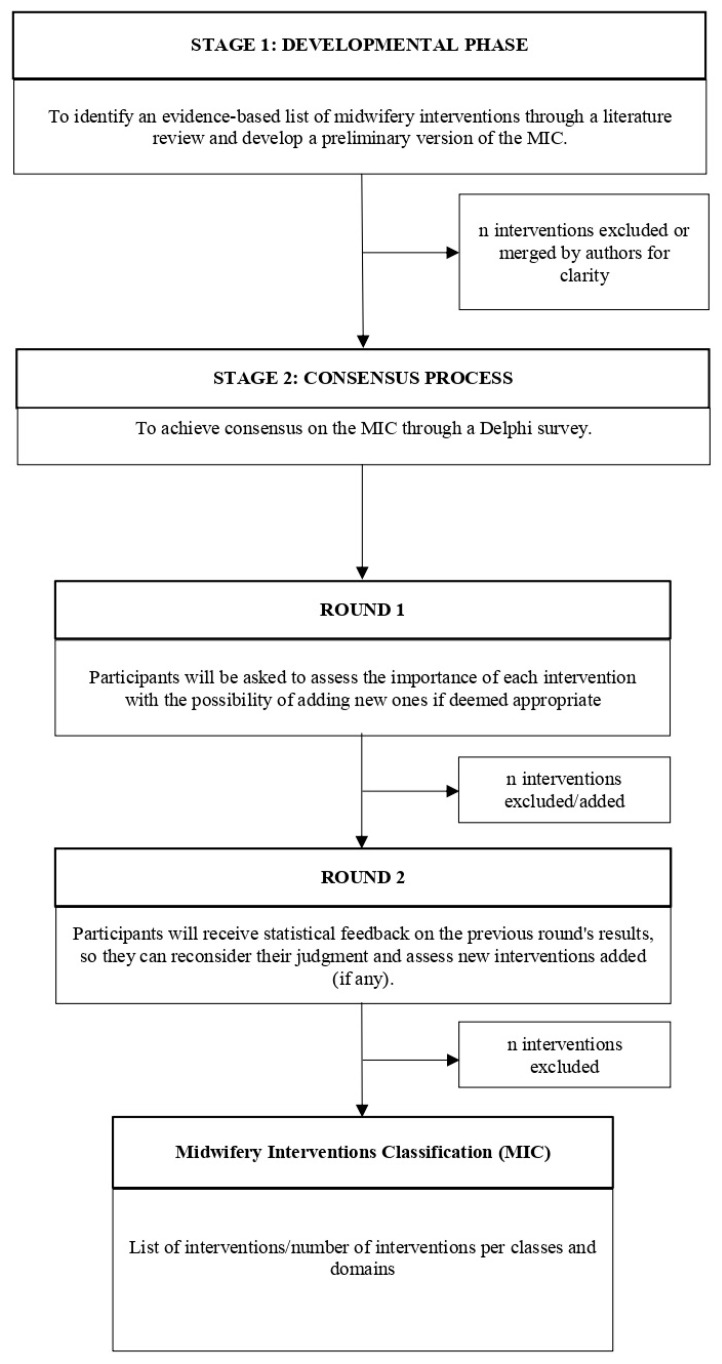
Flowchart of the study design.

**Figure 2 healthcare-11-00919-f002:**
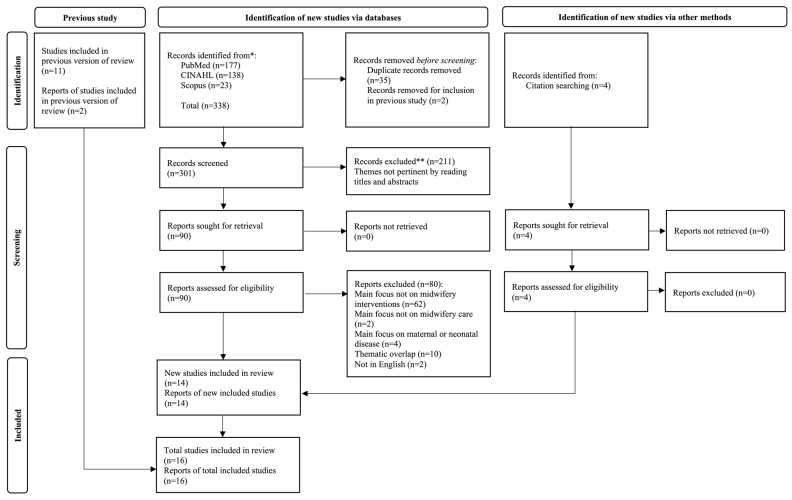
Selection flow diagram. Legend: * indicates the first identification phase; ** indicates excluded records.

**Figure 3 healthcare-11-00919-f003:**
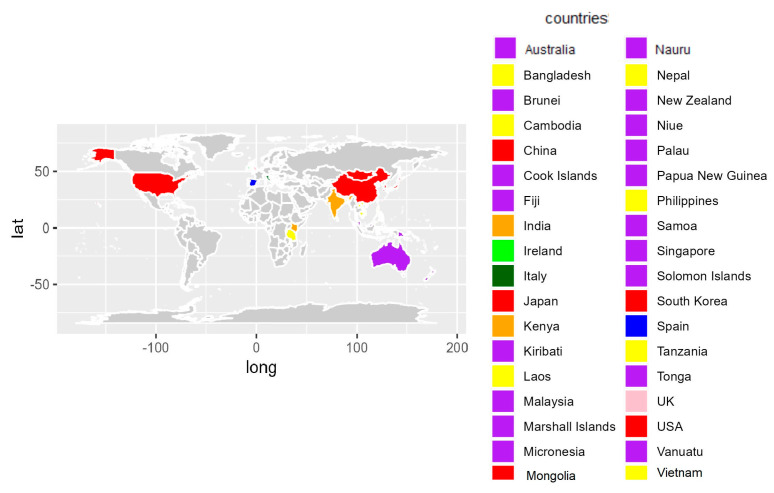
Distribution of the countries where the studies were performed.

**Table 1 healthcare-11-00919-t001:** Literature search strategy.

Databases	Search Queries	Studies
	(Last Updated on 15 February 2023)	(n)
PubMed	Search (((“Maternal Health”[Mesh] or Health, Maternal)) AND (“Midwifery”[Mesh] or Midwives or midwife)) AND (“Quality Indicators, Health Care”[Mesh]OR Quality Indicators, Healthcare OR Healthcare Quality Indicator) Sort by: Best Match	177
CINAHL	AB (midwifery or midwives or midwife) AND AB maternal health AND AB (patient outcomes or quality of care)	138
Scopus	TITLE-ABS-KEY ((maternal AND health OR health, AND maternal) AND (midwifery OR midwives) AND (quality AND indicators, AND health AND care OR quality AND indicators, AND healthcare OR healthcare AND quality AND indicator))	23

**Table 2 healthcare-11-00919-t002:** Summary of the proposed MIC.

Midwifery Interventions		References															
		Alliman et al., 2019 [41]	Blanc et al., 2016 [40]	Butcher et al., 2020 [33]	Day et al., 2021 [32]	Devane et al., 2019 [39]	Diamond-Smith et al., 2022 [31]	Escuriet et al., 2015 [38]	Flenady et al., 2016 [30]	Lavender, 2016 [37]	Lazzaretto et al., 2018 [2]	Nababan et al., 2017 [29]	Obara et al., 2014 [28]	Pricilla et al., 2017 [27]	Saturno-Hernàndez et al., 2019 [26]	Simpson et al., 2019 [35]	Ueda et al., 2019 [34]
Italian Wording	English Wording																
Accoglienza in una struttura sanitaria o sociale	Reception in a healthcare or social facility			x													
Alimentazione con biberon	Bottle feeding			x													
Alimentazione con tazza: neonato	Cup feeding: newborn			x													
Alimentazione del neonato	Infant feeding			x		x						x					
Alimentazione della donna	Maternal feeding		x	x							x					x	
Allattamento al seno: avvio	Breastfeeding initiation		x		x				x	x		x	x			x	x
Allattamento al seno: mantenimento	Breastfeeding continuation																x
Ascolto attivo	Active listening			x													
Assistenza al I stadio del travaglio	First stage of labor care attendance										x						x
Assistenza al II stadio del travaglio	Second stage of labor care attendance										x						x
Assistenza al III stadio del travaglio	Third stage of labor care attendance									x	x	x					
Assistenza al IV stadio del travaglio	Fourth stage of labor care attendance			x		x		x									
Assistenza al parto	Birth attendance			x				x			x		x				
Assistenza all’adattamento neonatale	Neonatal adaptation care		x	x		x			x	x		x	x				
Assistenza allo stadio prodromico del travaglio	Early labor (latent phase) care attendance										x						x
Assistenza domiciliare	Home care attendance	x															
Assistenza ostetrica in gravidanza	Antenatal care (ANC) attendance			x													
Assistenza ostetrica in puerperio	Postnatal care attendance		x			x		x									
Attuazione di meccanismi di responsabilità sociale su un’assistenza alla maternità rispettosa e dignitosa	Implementation of mechanisms for social responsibility in Respectful Maternity Care (RMC)						x										
Cardiotocografia: gestione delle alterazioni	Cardiotocography: management of alterations					x								x		x	
Case management	Midwifery case management			x													
Clampaggio del cordone	Cord clamping										x		x				
Coinvolgimento del caregiver	Involvement of the caregiver		x								x	x					
Coinvolgimento di membri della comunità sull’assistenza alla maternità rispettosa e dignitosa	Involvement of community members in RMC						x										
Collaborazione con l’équipe	Collaboration with the team			x												x	
Condivisione di obiettivi	Sharing of objectives			x													
Consegne ostetriche	Midwifery clinical handover			x		x										x	
Consulenza	Midwifery clinical consultation			x													
Consulenza assistenziale multidisciplinare	Multi-disciplinary clinical consultation			x													
Consulenza telefonica	Telephone counseling			x													
Continuità assistenziale ostetrica (one to one)	Midwifery Led Care										x						
Contratto con l’assistito	Contract with the assisted person			x													
Controllo del carrello dell’emergenza	Emergency cart control			x													
Controllo delle infezioni	Infection control		x	x					x			x				x	
Counseling	Counseling	x		x													
Counseling alle donne sulla consapevolezza di un’assistenza alla maternità rispettosa e dignitosa	Counseling on women’s awareness of RMC						x										
Counseling sessuale	Sexual counseling			x													
Counseling: accudimento del neonato	Counseling: infant care			x													
Counseling: allattamento	Counseling: breastfeeding			x										x			
Counseling: comportamenti di salute	Counseling: health behaviors	x		x					x								x
Counseling: diagnosi prenatale	Counseling: prenatal diagnosis			x													
Counseling: eliminazione intestinale del neonato	Counseling: infant bowel elimination			x													
Counseling: eliminazione intestinale della donna	Counseling: maternal bowel elimination			x													
Counseling: eliminazione urinaria del neonato	Counseling: infant urinary elimination			x													
Counseling: eliminazione urinaria della donna	Counseling: maternal urinary elimination			x													
Counseling: nutrizione del neonato	Counseling: infant nutrition			x													
Counseling: nutrizione della donna	Counseling: maternal nutrition			x										x			
Counseling: pianificazione familiare	Counseling: family planning	x		x					x	x		x		x			
Counseling: segni e sintomi	Counseling: signs and symptoms													x		x	
Counseling: sicurezza del neonato	Counseling: infant safety			x													
Cura del cordone ombelicale	Umbilical cord care								x	x							
Cura del perineo	Perineal care			x												x	
Cura delle lacerazioni perineali	Perineal tears care			x							x						x
Documentazione	Clinical documentation			x		x					x	x	x			x	x
Documentazione del travaglio: partogramma	Labor clinical documentation: partograph			x		x					x	x	x			x	x
Ecografia office	Obstetric Point-of-Care Ultra-sound (POCUS)			x													
Educazione alla salute	Health education	x		x													
Eliminazione intestinale del neonato	Infant bowel movements			x													
Eliminazione intestinale della donna	Women’s bowel movements			x													
Eliminazione urinaria del neonato	Infant urinary elimination			x													
Eliminazione urinaria della donna	Women’s urinary elimination			x												x	
Esame obiettivo del neonato	Neonatal physical examination								x				x				
Esame obiettivo della donna	Women’s physical examination	x				x								x			
Facilitazione dell’apprendimento	Facilitation of learning			x													
Facilitazione delle visite	Facilitation of visits			x													
Follow-up telefonico	Telephone follow-up			x													
Formazione del personale sanitario su un’assistenza alla maternità rispettosa e dignitosa	Training healthcare staff on RMC						x										
Gestione degli approvvigionamenti	Management of supplies			x													
Gestione dei campioni di laboratorio	Management of laboratory samples			x													
Gestione dei codici di gravità	Management of severity codes			x													
Gestione dei farmaci	Management of medications			x													
Gestione del dolore	Pain management		x	x		x			x		x					x	
Gestione dell’allergia	Allergy management			x													
Gestione dell’ambiente	Environmental management			x													
Gestione dell’ambiente: benessere	Environmental management: well-being			x			x										
Gestione dell’ambiente: sicurezza	Environmental management: safety			x													
Gestione della tecnologia	Technology management			x													
Gestione delle profilassi del neonato	Management of neonatal prophylaxis								x				x				x
Gestione delle risore economiche	Management of economic resources			x													
Gestione delle vaccinazioni	Management of vaccinations			x					x				x				
Guida al sistema sanitario	Guide to the healthcare system			x													
Guida preventiva alle situazioni critiche	Preventive guidance for critical situations			x													
Identificazione dei rischi	Risk identification			x													
Identificazione della persona assistita	Identification of the assisted person			x													
Idroterapia	Hydrotherapy															x	
Igiene del neonato	Neonatal hygiene			x													
Igiene della donna	Women’s hygiene			x												x	
Implementazione di programmi di miglioramento della qualità dell’assistenza	Implementation of quality improvement programs						x										
Implementazione di regolamenti/raccomandazioni e linee guida a supporto di un’assistenza alla maternità rispettosa e dignitosa	Implementation of regulations/recommendations and guidelines to RMC						x										
Incannulazione venosa	Venous cannulation			x													
Interpretazione dei dati di laboratorio	Interpretation of laboratory data			x													
Massaggio	Massage			x													
Mediazione culturale	Cultural mediation			x													
Miglioramento del coping	Coping improvement			x													
Miglioramento dell’alfabetizzazione sulla salute	Health literacy improvement			x													
Miglioramento dell’autoefficacia	Self-efficacy improvement			x													
Miglioramento dell’autostima	Self-esteem improvement			x													
Miglioramento della collaborazione	Collaboration improvement			x								x				x	
Miglioramento della disponibilità all’apprendimento	Improved learning availability			x													
Miglioramento della socializzazione	Socialization improvement			x													
Monitoraggio dei parametri vitali	Vital signs monitoring		x	x										x		x	
Monitoraggio della politica sanitaria	Health policy monitoring			x													
Monitoraggio della qualità	Quality monitoring			x													
Osservazione post-partum	Postpartum observation		x								x	x					x
Personalizzazione dell’assistenza	Personalized care													x			
Pianificazione dell’assistenza	Care planning	x				x											x
Pianificazione della dimissione	Discharge planning	x		x		x						x					
Potenziamento del ruolo	Role empowerment			x													
Potenziamento della consapevolezza di sé	Self-awareness empowerment			x													
Precauzioni d’uso per il lattice	Precautions for latex use			x													
Prelievo ematico capillare	Capillary blood sampling			x													
Prelievo: campione di sangue venoso	Venous blood sample collection			x													
Preparazione al parto	Antenatal class			x										x			x
Prescrizione: test diagnostico	Prescription: diagnostic test			x													
Prescrizione: trattamento non farmacologico	Prescription: non-pharmacological treatment			x					x					x			
Presenza	Presence			x													
Prevenzione dell’emorragia post-partum	Prevention of postpartum hemorrhage		x	x	x				x	x	x	x				x	x
Prevenzione delle cadute	Fall prevention			x													
Promozione del movimento in travaglio	Promotion of movement during labor		x														
Promozione del ruolo genitoriale	Promotion of parental role	x		x													
Promozione dell’attaccamento genitore-bambino	Promotion of parent–child attachment		x	x													
Promozione dell’attività fisica	Promotion of physical activity			x													
Promozione dell’empowerment	Empowerment promotion			x										x			x
Promozione della cura di sé (self-care)	Self-care promotion			x													
Promozione della normalità della nascita	Promotion of normal labor and birth										x						
Promozione di posizioni materne libere al parto	Promotion of free maternal positions during childbirth		x								x						x
Promozione di un’assistenza alla maternità rispettosa e dignitosa (advocacy)	Promotion of respectful maternity care (advocacy)						x										
Promozione donazione sangue cordone ombelicale	Promotion of umbilical cord blood donation			x													
Raccolta dati ai fine di ricerca	Data collection for research purposes			x													
Regolazione della temperatura	Temperature regulation		x	x		x			x	x		x	x				
Relazione su un evento accidentale (incident reporting)	Reporting of accidental events (incident reporting)			x													
Rooming in	Rooming in		x														
Scambio di informazioni relative alla salute	Exchange of health information			x													
Screening	Screening	x		x					x					x	x		
Skin to skin	Skin-to-skin contact		x		x						x	x	x			x	x
Somministrazione di analgesici	Administration of analgesics			x													
Somministrazione di farmaci	Administration of medication			x		x										x	
Sorveglianza	Surveillance			x													
Sorveglianza: gravidanza a termine	Surveillance: term pregnancy			x													
Sorveglianza: teleassistenza	Surveillance: teleassistance			x													
Sostegno al processo decisionale	Support for decision-making process	x		x										x		x	
Sostegno del caregiver	Support for caregiver			x													
Sostegno emozionale	Emotional support			x												x	
Sostegno nella gestione del comportamento	Support in behavior management			x													
Sostegno nella modifica del comportamento	Support in behavior modification			x													
Supervisione del personale	Personnel supervision			x													
Supporto a chi fornisce un’assistenza alla maternità rispettosa e dignitosa	Support for those providing RMC						x										
Sutura del perineo	Perineal suturing			x							x						x
Sviluppo del personale	Staff development			x													
Sviluppo della salute della comunità	Community health development			x													
Sviluppo di programmi	Program development			x													
Trasferimento all’interno della struttura	Transfer within the facility			x													
Triage ostetrico	Obstetric triage			x													
Triage telefonico	Telephone triage			x													
Tutela dei diritti della persona assistita	Protection of the rights of the assisted person			x													
Tutorato: personale dipendente	Staff tutoring			x													
Tutorato: studenti	Student tutoring			x													
Valutazione benessere fetale: battito cardiaco fetale (BCF)	Fetal well-being evaluation: fetal heart rate (FHR)			x		x					x			x		x	
Valutazione benessere fetale: movimenti attivi fetali (MAF)	Fetal well-being evaluation: fetal movement (FM)													x		x	
Valutazione dei presidi	Assessment of equipment			x													
Valutazione del benessere emotivo	Emotional well-being assessment					x					x						x
Valutazione del rischio ostetrico feto/neonatale	Obstetric risk assessment: fetal/neonatal	x	x			x											x
Valutazione del rischio ostetrico materno	Obstetric risk assessment: maternal	x	x			x											x
Valutazione dell’attività contrattile uterina (ACU)	Assessment of uterine contractions										x					x	
Visualizzazione guidata	Guided visualization			x													

## Data Availability

Data will be available from the corresponding author upon reasonable request.

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
