# Peer review of "Developmental Strategy and Validation of the Midwifery Interventions Classification (MIC): A Delphi Study Protocol and Results from the Developmental Phase"

_healthcare, 2023, doi:10.3390/healthcare11060919_

Round 1

Reviewer 1 Report

The manuscript has a very interesting topic.

They want to develop a classification (code system) of midwifery interventions in Italy. The authors refer to the fact that there are already similar publications at the international level. They want to develop this for interventions performed by Italian midwives.

Keeping patient safety, education and research in mind.

The developed material will also serve to better recognize the work of midwives.

The manuscript can be seen as a project plan, or as a research outline.

For this, the authors describe everything exactly step by step.

The manuscript contains a figure that summarizes the study design, like a flowchart.

The manuscript does not contain statistical data/processing.

In point 2.2, the authors describe precisely the characteristics of the participants.

The abstract is formally recommended for revision:

- introduction

- purpose / aim

- methods

- discussion

- conclusion.

The manuscript cites 32 references, appropriately. 26 of them have DOI identifiers.

The nine authors work in seven institutes, two of which are in England and five in Italy.

The authors state at the end of the manuscript that further research will be needed for validation. The authors will also recommend the results of the research for international use.

The reviewer recommends revising the abstract.

The manuscript is then recommended for acceptance and publication.

Author Response

Dear reviewer, please find here attached the point-by-point responses. 

Reviewer 2 Report

This is a well written protocol for a Delphi study.  It will be relevant to others who may be  interested in undertaking this methodology.   I will also be interested in the results of the Delphi study when this becomes available.  The authors can be commended for the thoroughness of their approach.   

Author Response

(The authors gave the same response as above.)

Reviewer 3 Report

Thank you for the opportunity to review the article, "Developmental strategy and validation of the Midwifery Interventions Classification (MIC): a Delphi study protocol."

This is an important article that expertly describes a Delphi process for identifying midwifery quality indicators. It would be helpful if the authors could discuss the context for Italian midwifery to understand what the Delphi process means for midwives in Italy. 

Author Response

(The authors gave the same response as above.)

Reviewer 4 Report

It is an interesting study protocol, which aims to describe the developmental strategy to define a Midwifery Interventions Classification (MIC), an evidence based, standardized taxonomy and classification of midwifery interventions based on the Italian maternity care context.

In my opinion, it could be more valuable for scientific community to be presended the results and conclution of your final study and not only your study protocol. This manuscript does not provide a significant content which is worth to be published. 

Author Response

(The authors gave the same response as above.)
